# OpenReview forum: "Shake-It-Off: Jailbreaking Black-Box Large Language Models by Shaking Off Objectionable Semantics"
_ICLR.cc/2025/Conference — ICLR 2025 Conference Withdrawn Submission_

### Official Review · Reviewer_nrrX · 2024-11-01

**Soundness:** 2
**Presentation:** 3
**Contribution:** 2
**Rating:** 5
**Confidence:** 4

**Summary:**

This paper presents SHAKE (Shake-It-Off), a new efficient method for jailbreaking large language models. SHAKE iteratively modifies model outputs to bypass safeguards and generate objectionable content, using only the model's responses. Experiments show it outperforms baseline methods in attack success while requiring less time and model access. The study aims to understand these attacks from a red team perspective, likely to improve future LLM defenses.

**Strengths:**

1. The jailbreaking attack is an interesting topic.
2. The reported attack success rate appears to be notably effective, which underscores the importance of developing robust defense mechanisms for large language models.

**Weaknesses:**

1. The novelty of the proposed approach appears to be somewhat limited in the context of existing research.
2. Given that the authors present their attack as a black-box method, it's surprising that they didn't extend their evaluation to include prominent commercial models such as Claude and ChatGPT. This omission raises questions about the method's broader applicability.
3. The paper would benefit from a more comprehensive discussion of potential defense strategies. It would be valuable to see an evaluation of the proposed method against representative defense mechanisms, providing a more balanced perspective on the attack's effectiveness in real-world scenarios.

**Questions:**

Please see comments.

---

> ### Author Response · Authors · 2024-11-28
> **Response to Reviewer nrrX**
>
> We sincerely thank the reviewer for the valuable feedback. We will address the concerns in the following comments.
>
> > The novelty of the proposed approach appears to be somewhat limited in the context of existing research.
>
> As the major concerns on LLMs’ safety are around commercial (black-box) models, the research of black-box jailbreaking attack/defense is more practical than white-box attacks. Compared to existing black-box attacks (e.g., PAIR and ReNeLLM), our proposed method is more intuitive and efficient (as claimed in the statement of contribution, Lines 134-144). Other black-box attacks like PAIR and ReNeLLM mainly update the adversarial prompt by asking the attack LLM to revise the current adversarial prompt *without any guidance*. As demonstrated in Figure 2 (b), such revision is like randomly sampling around the adversarial prompt. **We propose an optimization paradigm to guide the sampling process.** The optimization paradigm is
> + Interpretable. As we have carefully discussed in Section 3, we construct a surrogate optimization objective (Equation (4)) that naturally extends the naive optimization objective (Equation (3)) of a jailbreaking attack.
> + Efficient. By guiding the sampling process, our method can further reduce the queries to the attack model and thus be more efficient than PAIR.
>
> > Given that the authors present their attack as a black-box method, it's surprising that they didn't extend their evaluation to include prominent commercial models such as Claude and ChatGPT. This omission raises questions about the method's broader applicability.
>
> We thank the reviewer for the constructive suggestion. We conduct auxiliary experiments on GPT-4o GPT-4o-mini during the rebuttal period. Our proposed method reaches **56% rechecked ASR** on the AdvBench-sub dataset, showing that SHAKE is also effective against more robust, modern models. Notice that the reported **rechecked ASRs on GPT-4 of PAIR and ReNeLLM are 48% and 58.9%**. Considering that the GPT-4o is more recently released, we claim that the performance of SHAKE is at least comparable to the popular baseline.

---

> > ### Comment · Reviewer_nrrX · 2024-12-02
> > **Reply to authors**
> >
> > Thank you for your response. I maintain my current rating score.

---

### Official Review · Reviewer_8Edi · 2024-11-03

**Soundness:** 3
**Presentation:** 3
**Contribution:** 3
**Rating:** 5
**Confidence:** 4

**Summary:**

The paper presents a new jailbreaking attack against LLMs in the black box setting. Borrowing techniques from PAIR and HSJ attacks, the attack proceeds in multiple steps: remove sensitive words/phrases from a query, construct an initial adversarial prompt, and iterate while the prompt is rejected by the model. Each iteration includes sampling of prompts and keeping the ones closest to the decontaminated one in the embedding space. The paper evaluates the attack against llama 2 and vicuna models and compares the performance against three baselines: autoDAN, GCG, and ReNeLLM. The evaluation shows higher success with higher efficiency.

**Strengths:**

The paper describes the proposed attack clearly and also provides the algorithm and walks the reader through the algorithm steps. Furthermore, the appendix shows the prompts used for sampling and shows the sensitive words/phrases used for decontamination. The paper also includes a threat model, which contextualizes the proposed attack.

The attack approach is computationally efficient because the sampling approach is guided by similarity to a decontaminated prompt, potentially resulting in lower query complexity (to the model under attack).

The evaluation shows that the proposed attack performs better than white box or black box baselines.

**Weaknesses:**

The paper needs to present a clearer argument regarding the contributions over the PAIR attack. It appears as if the paper borrows several techniques from the PAIR attacks, but adds a sampling step so that the adversarial sample is closer to a benign sample. The evaluation section does not show if the proposed attack improves over PAIR. Most probably, the authors forgot to add those results to the evaluation section because the introduction to that section mentions PAIR as a baseline. As such, the evaluation section needs to compare against the PAIR attack and provide an ablation study showing that the added sampling step improves adversarial success and improves efficiency by reporting queries to the model under attack.

Moreover, it would be useful to the reader to see if the attack works against commercial models, such ChatGPT or other large models such as Mistral or Llama 3. The evaluation is performed against models that are considered by today's standards to be smaller and older. The PAIS paper, for example, shows evaluation over GPT, Gemini, and Claude. It should be feasible to run the attack against such models.

It would have also been useful to show examples of the generated adversarial prompts.

The paper does not include an ethics discussion, nor does it discuss potential defenses or mitigation. It is important for the paper to have a discussion section about these issues especially as it generates harmful prompts for aligned language models.

**Questions:**

How does the performance of the proposed attack compare against PAIR on commercial models?

**Details Of Ethics Concerns:**

The paper does not discuss the ethical implications for an attack against LLMs (generating harmful content).

---

> ### Author Response · Authors · 2024-11-28
> **Response to Reviewer 8Edi**
>
> We sincerely thank the reviewer for the valuable feedback. We will address the concerns in the following comments.
>
> ### Weakness
>
> > The paper needs to present a clearer argument regarding the contributions over the PAIR attack. It appears as if the paper borrows several techniques from the PAIR attacks, but adds a sampling step so that the adversarial sample is closer to a benign sample.
>
> As mentioned in Lines 45-46, and 167-168, our proposed method is **more interpretable and efficient** compared to PAIR. Our contributions over PAIR can be summarized as follows. PAIR updates the adversarial prompt by asking the attack LLM to revise the current adversarial prompt without any guidance. As demonstrated in Figure 2 (b), such revision is like randomly sampling around the adversarial prompt. **Our method does not “add a sampling step” to PAIR. Instead, we propose an optimization paradigm to guide the sampling process.** The optimization paradigm is
> + Interpretable. As we have carefully discussed in Section 3, we construct a surrogate optimization objective (Equation (4)) that naturally extends the naive optimization objective (Equation (3)) of a jailbreaking attack.
> + Efficient. By guiding the sampling process, our method can further reduce the queries to the attack model and thus be more efficient than PAIR.
>
> As for evaluation, we compare our method with PAIR in Table 1. Our proposed method is significantly more effective than the reported rechecked ASR of PAIR. Besides, we also conduct auxiliary experiments on GPT-4o and GPT-4o-mini during the rebuttal period. We refer the reviewer to the Questions part for more detail.
>
> ### Questions
>
> > How does the performance of the proposed attack compare against PAIR on commercial models?
>
> We conduct auxiliary experiments on GPT-4o and GPT-4o-mini during the rebuttal period. Our proposed method reaches **56% rechecked ASR** on the AdvBench-sub dataset, showing that SHAKE is also effective against more robust, modern models. Notice that the reported rechecked ASRs on GPT-4 of PAIR and ReNeLLM are 48% and 58.9%. Considering that GPT-4o and GPT-4o-mini are more recently released LLMs, we claim that the performance of SHAKE is at least comparable to the popular baseline.

---

### Official Review · Reviewer_yteN · 2024-11-03

**Soundness:** 3
**Presentation:** 3
**Contribution:** 3
**Rating:** 6
**Confidence:** 4

**Summary:**

The authors propose a novel black-box jailbreaking strategy, SHAKE, which iteratively adjusts the target query in the direction of  a decontaminated version, gradually removing objectionable semantics and ultimately bypassing the model's safeguards.

**Strengths:**

1. Simple but effective strategy for jailbreaking.
2. Paper is well written.
3. Comprehensive comparison with prior work.

**Weaknesses:**

1.The evaluation section has very limited experiments.

2. Lack insights on why the attack works. See questions below.

**Questions:**

1. Your comparison is currently limited to prior work on smaller LLMs. It would be valuable to see how your attack performs on larger models, such as GPT-4, Claude, or Gemini, given that your attack is black-box in nature. Additionally, comparing your method with other black-box attacks, like ReNeLLM and PAIR, under these conditions would improve the evaluation.

2. I didn't see a clear reasoning for why the decontaminated query is constructed by replacing objectionable semantics with [PAD] tokens. Why not use other tokens, or even allow the target LLM to decontaminate the query itself to make it less objectionable? An ablation study on this approach could provide better insights into why the attack is effective.

3. Can you provide more insight on why other black-box attacks like PAIR and ReNeLLM don't work as well given there is significant overlap between those strategies and yours except the fact that you move in the direction of  the less objectionable query per step. I think results for Question 2 may provide answers to this as well.

---

> ### Author Response · Authors · 2024-11-28
> **Response to Reviewer yteN**
>
> We sincerely thank the reviewer for the valuable feedback. We will address the concerns in the following comments.
>
> ### Weakness 1
>
> > The evaluation section has very limited experiments.
>
> As a quickly developing area of research, there is a lack of popular acknowledged baselines for jailbreaking attacks. To the best of our knowledge, the four methods (i.e., GCG, AutoDAN, PAIR, and ReNeLLM) compared by our paper are the most widely used baseline in concurrent works. As shown in Tables 1 and 3, our method outperforms these baselines in AdvBench and AdvBench-Sub datasets.
>
> To further improve validity, we conduct auxiliary experiments on GPT-4o and GPT-4o-mini during the rebuttal period. Our proposed method reaches **56% rechecked ASR** on the AdvBench-sub dataset, showing that SHAKE is also effective against larger models. Notice that the reported **rechecked ASRs on GPT-4 of PAIR and ReNeLLM are 48% and 58.9%**. Considering that the GPT-4o and GPT-4o-mini (and thus more powerful, as reported in the LMSYS leaderboard (https://lmarena.ai/)) are more recently released, we claim that the performance of SHAKE is at least comparable to the popular baseline.
>
> ### Weakness 2
>
> > Lack insights on why the attack works. See questions below.
>
> We will reply to this issue in Question 3.
>
> ### Question 1
>
> > Your comparison is currently limited to prior work on smaller LLMs. It would be valuable to see how your attack performs on larger models, such as GPT-4, Claude, or Gemini, given that your attack is black-box in nature. Additionally, comparing your method with other black-box attacks, like ReNeLLM and PAIR, under these conditions would improve the evaluation.
>
> Thanks for your constructive suggestion. We have conducted auxiliary experiments on GPT-4o and GPT-4o-mini to improve the evaluation as mentioned in the Weakness part.
>
> ### Question 2
>
> > I didn't see a clear reasoning for why the decontaminated query is constructed by replacing objectionable semantics with [PAD] tokens. Why not use other tokens, or even allow the target LLM to decontaminate the query itself to make it less objectionable? An ablation study on this approach could provide better insights into why the attack is effective.
>
> The intuition of our decontamination process is to replace the objectionable tokens in the sentence with some “absolute neutral” tokens. In some LLMs, the token [PAD] (after peeling off the punctuations) is tokenized to 0 so that it is not similar to any other token. That is why we choose [PAD].
>
> We sincerely thank the reviewer for providing such insightful comments. We will add an ablation study by using other tokens and auto-decontaminating by LLMs in future revisions.
>
> ### Question 3
>
> > Can you provide more insight on why other black-box attacks like PAIR and ReNeLLM don't work as well given there is significant overlap between those strategies and yours except the fact that you move in the direction of the less objectionable query per step. I think results for Question 2 may provide answers to this as well.
>
> Sure! Here is a summary of the insight. Other black-box attacks like PAIR and ReNeLLM mainly update the adversarial prompt by asking the attack LLM to revise the current adversarial prompt without any guidance. As demonstrated in Figure 2 (b), such revision is like randomly sampling around the adversarial prompt. Our method **pointed out a “less objectionable direction” to guide the sampling process** so that the updated adversarial prompt would be more benign according to some similarity metrics.

---

> > ### Comment · Reviewer_yteN · 2024-11-29
> >
> > Thank you for your response. I would like to maintain my positive score as I like the idea of the paper but needs some more ablations to better understand the proposed attack. Some possible ablations could be,
> >
> > i) Why does the attack work better than baselines on smaller sized models versus but does not outperform them for large-sized models.
> >
> > ii) Ablation on Question 2 as mentioned before.

---

### Official Review · Reviewer_tpgY · 2024-11-04

**Soundness:** 2
**Presentation:** 2
**Contribution:** 1
**Rating:** 3
**Confidence:** 5

**Summary:**

The paper introduces SHAKE, a black-box jailbreaking attack method for LLMs. SHAKE iteratively removes objectionable semantics from adversarial prompts, leading to successful bypassing of model safeguards. The authors validate SHAKE's effectiveness and efficiency on various datasets and LLMs, showing improvements in attack success rates and reduced running time compared to existing methods.

**Strengths:**

The method is clearly explained, and the writing is well-executed.

**Weaknesses:**

All of the evaluated LLMs, such as Vicuna and LLaMA2, are open-weight models that have already been shown to be quite fragile, casting doubt on how effective the attack would be against more robust, modern models. Furthermore, the method demonstrates only a marginal improvement over the current state-of-the-art.

I don't believe this type of "yet another jailbreak" paper provides significant value to the research community.

**Questions:**

Q1: All of the evaluated LLMs, such as Vicuna and LLaMA2, are already shown to be quite fragile against jailbreaks. Pushing the ASR on these models from ~90% to 100% is not a novel contribution. It's unclear how effective the attack would be on more robust, modern models.

Q2: How does the attack perform on proprietary models? The attack is described as a black-box method, but none of the evaluated models seem to be true black-box models.

---

> ### Author Response · Authors · 2024-11-28
> **Response to Reviewer tpgY**
>
> We sincerely thank the reviewer for the valuable feedback. We will address the concerns in the following comments.
>
> ### Weakness
>
> > All of the evaluated LLMs, such as Vicuna and LLaMA2, are open-weight models that have already been shown to be quite fragile, casting doubt on how effective the attack would be against more robust, modern models.
>
> During the rebuttal period, we conduct auxiliary experiments on GPT-4o and GPT-4o-mini, both of which are modern and robust models that outperform the popular baseline GPT-4 in the LMSYS leaderboard (https://lmarena.ai/). **Our proposed method reaches 56% rechecked ASR on the AdvBench-sub dataset against both GPT-4o and GPT-4o-mini**, showing that SHAKE is also effective against more robust, modern models. **Notice that the reported rechecked ASRs on GPT-4 of PAIR and ReNeLLM are 48% and 58.9%**, respectively. Considering that GPT-4o and GPT-4o-mini are more powerful LLMs, we claim that the performance of SHAKE is at least comparable to the popular baseline.
>
> > Furthermore, the method demonstrates only a marginal improvement over the current state-of-the-art.
>
> As claimed in Lines 45-46, 167-168, our proposed method is more **interpretable and efficient** than the current SOTA. Motivated by the zeroth-order optimization framework (cf. Lines 175-190), SHAKE introduces a surrogate optimization objective (Equation (4)) to the original intractable optimization problem. As demonstrated in Figure 2 (b), the surrogate optimization objective selects the adversarial prompt most similar to the decontaminated objectionable query $T_{dec}$. This selection process would also speed up the jailbreaking attack by guiding the updating directions of the adversarial prompt.
>
> ### Question 1
>
> > All of the evaluated LLMs, such as Vicuna and LLaMA2, are already shown to be quite fragile against jailbreaks. Pushing the ASR on these models from ~90% to 100% is not a novel contribution. It's unclear how effective the attack would be on more robust, modern models.
>
> As claimed above, our method is also effective against GPT-4o and GPT-4o-mini, the robustness of which has been less studied. Apart from improving the ASR, our contribution also includes the novel interpretation of the jailbreaking attack process.
>
> ### Question 2
>
> > How does the attack perform on proprietary models? The attack is described as a black-box method, but none of the evaluated models seem to be true black-box models.
>
> We apologize for the possible confusion. We have added experiments on modern proprietary models as mentioned in the Weakness part. Besides, the proposed method only requires API access to the Llama and Vicuna LLMs. We choose these models for easy comparison to the existing baselines.

---

> > ### Comment · Reviewer_tpgY · 2024-12-02
> >
> > I appreciate the authors' efforts in conducting additional experiments on GPT-4o and 4o-mini. However, as noted by the authors, the proposed method's results are only comparable to, but not substantially better than, existing attacks. This undermines the paper's contribution as a work proposing a new jailbreak attack.
> >
> > Additionally, the new experiments do not expand the pool of open-weight models evaluated—the tested models remain limited to the very **outdated** Vicuna and Llama 2.
> >
> > I believe the paper requires substantial revisions before it is ready for publication and have decided to maintain my score.

---

### Note · Authors · 2024-12-03

I have read and agree with the venue's withdrawal policy on behalf of myself and my co-authors.